# Atomically precise copper dopants in metal clusters boost up stability, fluorescence, and photocatalytic activity

Yifei Zhang[1,2,5], Jingjing Zhang[2,3,5], Zhiwen Li[2], Zhaoxian Qin[2,3 ✉], Sachil Sharma[2,4] & Gao Li [2,3 ✉]

The structurally precise alloy nanoclusters have been emerged as a burgeoning nanomaterial for their unique physical/chemical features. We here report a rod-like nanocluster [$Au_{12}Cu_{13}(PPh_3)_{10}I_7$]($SbF_6$)$_2$ ($Au_{12}Cu_{13}$), which was generated through a transformation of a [$Au_9(PPh_3)_8$]$^{3+}$ intermediate in the presence of CuI, unveiled by time-dependent UV-vis spectroscopy, electrospray ionization mass spectrometry as well as single crystal X-ray diffraction. $Au_{12}Cu_{13}$ is comprised of two pentagonal bipyramids $Au_6Cu$ units and a pentagonal prism $Cu_{11}$ unit, where the copper and gold species are presented in +1 and 0 chemical states. The Cu-dopants significantly improved the stability and fluorescence (quantum yield: ~34%, 34-folds of homo-$Au_{25}(PPh_3)_{10}Br_7$). The high stability of $Au_{12}Cu_{13}$ is attributed to the high binding energy of iodine ligands, Au-Cu synergistic effects and its 16-electon system as an 8-electron superatom dimer. Finally, the robust $Au_{12}Cu_{13}$ exhibited high catalytic activity (~92% conversion and ~84% methyl formate-selectivity) and good durability in methanol photo-oxidation.

[1] Institute of Catalysis for Energy and Environment, College of Chemistry and Chemical Engineering Shenyang Normal University, Shenyang 110034, China. [2] State Key Laboratory of Catalysis, Dalian Institute of Chemical Physics, Chinese Academy of Sciences, Dalian 116023, China. [3] University of Chinese Academy of Sciences, Beijing 100049, China. [4] School of Advanced Sciences, Department of Chemistry, Vellore Institute of Technology, Andhra Pradesh (VIT-AP university), Amaravati, Andhra Pradesh 522237, India. [5] These authors contributed equally: Yifei Zhang, Jingjing Zhang. ✉email: qinzhaoxian@yeah.net; gaoli@dicp.ac.cn

In the famous lecture in 1959 with the title "*There is plenty of room at the bottom*", R. Feynman predicted the beginning of the research at the atomic level[1]. His prediction has been come true in a real sense with the emergence of research in the area of magic nanoclusters, where we can make nanoscale things at the atomic level in a highly controlled fashion[2]. Such functionalized nanoscale materials at the atomic level actually gained significant importance in technology progression due to their tunable physical and chemical properties[3,4]. Serving as an important part of such nanomaterials, the metal nanoclusters "$M_nL_m$", where $n$ and $m$ represent the number of metal atoms and surficial protecting ligands (L), respectively, have been highly promising in both fundamental research and practical applications, such as sensing[5], photo-luminescence[6], catalysis[4,7,8] and bio-imaging[9,10], owing to their unique and modulating physicochemical properties.

Metal atom doping is an important strategy to tune the structure and properties of metal nanoclusters[11–15]. Recently, the copper species doped into gold nanoclusters to generate Au-Cu alloy composites can well tailor their electronic structures and in turn to boosting the intrinsically physical and chemical properties, especially in luminescence and catalysis[13,16–21]. For example, Jin et al. have prepared an Au@Cu alloy nanocluster with a 71.4% quantity yield (QY) of fluorescence[18]. We found that the copper atoms doped in $M_{25}(SC_2H_4Ph)_{18}$ clusters, primarily enhanced the benzaldehyde-selectivity in the oxidation of styrene[20]. Unfortunately, Au-Cu alloy nanoclusters tend to show less stable under external environments[21–23] (e.g., light irradiation, oxidizer, and thermal conditions) due to the nature of $Cu^I$ and $Cu^0$ species, corroborated by the basis of Jellium Model[12]. Therefore, copper clusters and Au-Cu alloy clusters were often acquired under extreme conditions and stored in cool and dark place filled with inert atmosphere. The poor stability of Cu-based clusters becomes a burning issue and is still a big challenge for their applications.

Focused on the tunable properties and the improvement of stability of Cu doped gold clusters, we, herein, designed a strategy to prepare the rod-like $[Au_{12}Cu_{13}(Ph_3P)_{10}I_7](SbF_6)_2$ (abbreviated as $Au_{12}Cu_{13}$, hereafter) alloy cluster based on the self-assemble behavior of $[Au_9(PPh_3)_8](SbF_6)_3$ ($Au_9$ in short) clusters in the presence of CuI, which was further monitored by a serious of time-dependent measures including ultraviolet-visible (UV-vis) spectroscopy, electrospray ionization mass spectrometry (ESI-MS), and single crystal X-ray diffraction (SCXRD) to conclude the assembling process/mechanism of $Au_{12}Cu_{13}$ nanocluster. To our surprised, although $Au_{12}Cu_{13}$ cluster shares similar structure with the rod-like $[Au_{13}Ag_{12}(PPh_3)_{10}Cl_8](SbF_6)$[14] and $[Au_{25}(PPh_3)_{10}Br_7](SbF_6)_2$ (noted as $Au_{25}$ hereafter) clusters obtained in similar way, the dopant of Cu atoms in this alloy cluster has boosted its stability and photoluminescence significantly. The synergistic effects of Au and Cu atoms and strong Cu-I bonds may be the reason for the strong and high fluorescence of $Au_{12}Cu_{13}$ clusters. Finally, the $Au_{12}Cu_{13}$ cluster-based photo-catalyst was prepared based on the high stability and unique photo-properties, which shown excellent catalytic activity in the selective photo-oxidation of methanol.

## Results

**Synthesis and structure determination.** The alloy nanoclusters with composition of $[Au_{12}Cu_{13}(Ph_3P)_{10}I_7](SbF_6)_2$ were prepared in a step-by-step synthetic strategy. Briefly, a clear solution obtained upon mixing $Ph_3PAuCl$ and $AgSbF_6$ was reduced by $NaBH_4$ giving a dark-brown mixture, which was further reacted with CuI to produce the target clusters. To figure out the progress of alloy cluster formation, we firstly monitored the reaction solution by UV-vis spectroscopy and ESI-MS before the CuI addition. The obtained red-brown mixture gave five obvious peaks at 315, 350, 378, 443, and 511 nm in UV-vis spectrum (Fig. 1a), similar to that of the well-known $[Au_9(PPh_3)_8]^{3+}$ clusters[24]. And four main mass peaks at 1290.45 Da, 1705.90 Da, 1822.5 Da, and 2054.49 Da in the scale of $m/z$ 200–10,000 Da, assigned to the compositions of $[Au_9(PPh_3)_8]^{3+}$ ($m/z$ ($z=3$) 1290.33 Da calcd.), $[Au_8(PPh_3)_7]^{2+}$ ($m/z$ ($z=2$) 1705.85 Da calcd.), $[Au_9(PPh_3)_7Cl]^{2+}$ ($m/z$ ($z=2$) 1822.3 Da calcd.), and $[Au_9(PPh_3)_8SbF_6]^{2+}$ ($m/z$ ($z=2$) 2054.37 Da calcd.), respectively, were detected in positive ion mode of ESI-MS (Fig. 1b and Supplementary Figure 1), which confirmed the existence of

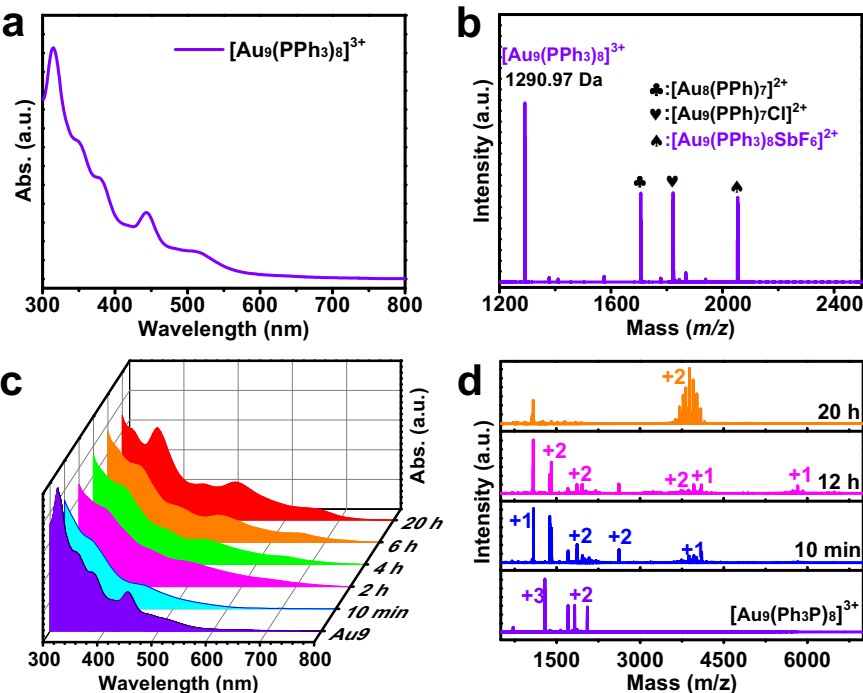

**Fig. 1 The conversion of Au₉ to Au₁₂Cu₁₃ clusters. a** UV-vis spectrum (in $CH_2Cl_2$) and **b** positive ion mode ESI-MS of $Au_9$ cluster. Conversion of $Au_9$ to $Au_{12}Cu_{13}$ clusters monitored by **c** time-dependent UV-vis spectra and **d** time-dependent ESI-MS.

$[Au_9(PPh_3)_8]^{3+}$ clusters in the obtained mixture. Note that such fragmentations are usual owing to the dissociation during ESI ionization of these phosphine protected metal nanoclusters[14]. All these results suggested that $Au_9$ cluster is dominant and served as starting material/precursor for the further construction of $Au_{12}Cu_{13}$ cluster.

Next time-dependent UV-vis spectroscopy (TD-UV-vis) and ESI-MS were applied to monitor the construction process of $Au_{12}Cu_{13}$ cluster originated from the reaction of $Au_9$ cluster and CuI in solution. As presented in Fig. 1c, the intensity of characteristic absorption peaks of $Au_9$ cluster at 315 and 443 nm decreased within initial 10 min, accompanied with the color deepening after CuI addition. Further, the peaks at 352, 434, 508, and 655 nm increased gradually in TD-UV-vis, indicating that the $Au_9$ cluster was transforming into other clusters. Figure 1d showed that the mass signals corresponding to $Au_9$ cluster disappeared quickly, and a series of new peaks (Supplementary Table 1) were shown up, demonstrating that $Au_9$ cluster reacted with CuI rapidly generating some metastable clusters, such as $[AuCu(Ph_3P)_2I]^+$, $[Au_3(Ph_3P)_3CuI]^+$, $[Au_6(Ph_3P)_6]^+$, $[Au_8Cu(Ph_3P)_8I]^{2+}$ and $[Au_9Cu(PPh_3)_8I_2]^{2+}$. These metastable clusters were further aggregated with each other to furnish $Au_{25-x}Cu_x$ clusters during the time-consuming thermodynamic process. Similar to the silver-doped $Ag_xAu_{25-x}$ nanoclusters[15], the number of Cu dopants in $M_{25}$ cluster ranges from 1 to 9, which is lower than that of the final Cu atoms (13) in the tested crystal sample. It may indicate that the Cu-doping process keeps going on during the crystallization process and the $Au_{12}Cu_{13}$ should be the most robust one in the final products[14].

The composition and purity of the obtained $Au_{12}Cu_{13}$ crystals were manifested by ESI-MS and X-ray photoelectron spectroscopy (XPS). As shown in Fig. 2a, the ESI-MS spectrum exhibited an intense peak at $m/z$ ~3350.45 Da (z = 2, calcd. $m/z$ ~3350.44 Da for $Au_{12}Cu_{13}P_{10}C_{180}H_{150}I_7$, deviation: 0.01 Da) in

the scale from $m/z$ ~1400 to 7000, corresponding to $[Au_{12}Cu_{13}(P-Ph_3)_{10}I_7]^{2+}$ cluster with high molecular purity. The isotopic pattern was found to be in exact agreement with simulated one and the peaks separation of $m/z$ ~ 0.5 Da, confirming the +2 charged state of $Au_{12}Cu_{13}$ nanocluster. These results demonstrated that the cluster specie in crystals is pure $[Au_{12}Cu_{13}(P-Ph_3)_{10}I_7]^{2+}$ instead of Au/Cu alternation and the $Au_{12}Cu_{13}$ cluster exhibit a 16-electron system (i.e. 25 (metal atoms) − 7 (I atoms) − 2 (charge) = 16).

The bimetal nature of $Au_{12}Cu_{13}$ cluster was also characterized by the wide scan XPS, Fig. 2b. The binding energy (BE) of Au $4f_{7/2}$ in $Au_{12}Cu_{13}$ can be deconvoluted to $Au^0$ (highlighted in orange) and $Au^I$ species[25] (Fig. 2c). Taking note of previous investigations, the final-state hole-shielding effect arising from extra atomic relaxation acts in opposite to electron donating effect of phosphine ligands, resulting in the increase of binding energy[26–28]. An unsymmetrical BE peak of Cu $2p_{3/2}$ near 933.1 eV was presented in Cu 2p XPS (Fig. 2d), excluding the existence of $Cu^{II}$ species (934.2 eV for Cu 2p in $Cu^{II}$)[29]. And the Cu $2p_{3/2}$ peak also can be deconvoluted into two peaks, corresponding to $Cu^0$ species at 932.5 eV and $Cu^I$ at 933.1 eV[30]. For validation, Auger electron spectrum showed a broad peak at 571.0 eV with a 1.0 eV positive shift with respect to the value of $Cu^I$ (570.0 eV) and a weak peak to $Cu^0$ at 568.0 eV (Supplementary Figure 2), indicating the presence of both $Cu^+$ and $Cu^0$ species in $Au_{12}Cu_{13}$. And the $Cu^0$ species, generally, should be attributed to the shared vertex Cu atom, while the $Cu^I$ one should belong to the peripheral and end vertex Cu atoms bonded directly with I atoms according to SCXRD analysis (vide infra). To the best of our knowledge, the $Au_{12}Cu_{13}$ cluster in present work is the first case of AuCu alloy cluster, where the Cu species exists both in +1 and 0 of chemical states, without any partial occupancy.

Based on the results of ESI-MS and UV-vis, we put forward explanation that the $Au_{12}Cu_{13}$ cluster should be prepared via the

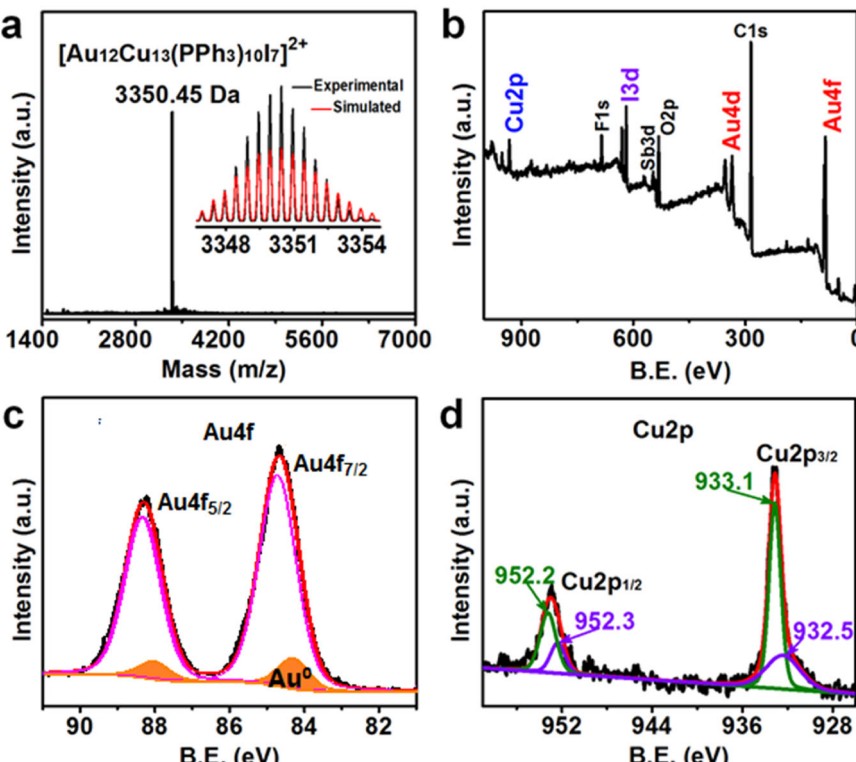

**Fig. 2 Physical property of the $Au_{12}Cu_{13}$ nanocluster. a** ESI-MS with isotopic pattern (insert). **b** Wide scan XPS. **c** Au 4 f and **d** Cu 2p XPS spectra of $Au_{12}Cu_{13}$ nanocluster.

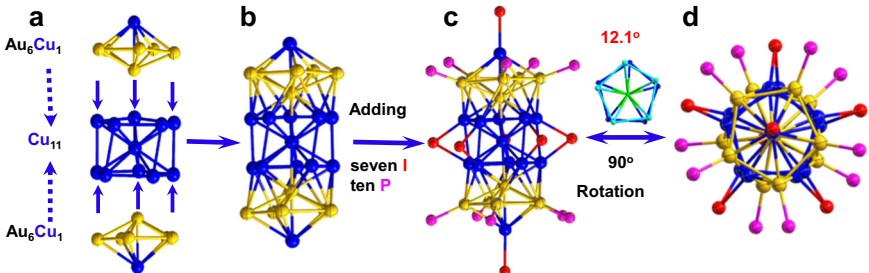

**Fig. 3 Configuration of Au$_{12}$Cu$_{13}$ cluster. a, b** Anatomy of Au$_{12}$Cu$_{13}$ cluster. **c** Side view and **d** top view. Color code: Au, yellow; Cu, blue and cyan; I, red; P, pink. Note that the other atoms are omitted for clarity.

transformation of Au$_9$ clusters upon a reaction with CuI. Thus, we conceived a sequence of experiments for the transformation of Au$_9$ clusters into another M$_{25}$ clusters, including rod-like Au$_{25-y}$Ag$_y$(PPh$_3$)$_{10}$Cl$_7$ (Au$_{25-y}$Ag$_y$) and homo Au$_{25}$(PPh$_3$)$_{10}$Br$_7$ (Au$_{25}$) in the presence of AgCl and KBr, respectively. Further, with respect to the self-assembly of these clusters, their structural analyses are of the essence. Therefore, the crystals of Au$_{12}$Cu$_{13}$ and Au$_{25}$ were grown by slow vapor diffusion of diethyl ether into a CH$_2$Cl$_2$ solution of clusters at ~10 °C over two weeks and characterized by SCXRD. Of note, the Au$_{25-y}$Ag$_y$ nanoclusters have been extensively studied[14,31].

SCXRD analysis revealed that Au$_{12}$Cu$_{13}$ and Au$_{25}$ clusters crystallize in *P21/n* and *P21/m* space groups, respectively. They share similar rod-like framework, as illustrated in Supplementary Figure 3. The metal core of Au$_{12}$Cu$_{13}$ can be divided into one Cu$_{11}$ and two Au$_6$Cu$_1$ units (Fig. 3a). Two Au$_6$Cu$_1$ units bind with the Cu$_{11}$ unit through Au-Cu metal bonds, generating an Au$_{12}$Cu$_{13}$ framework (Fig. 3b). The obtained Au$_{12}$Cu$_{13}$ metal kernel/core is directly ligated by ten PPh$_3$ ligands through Au-P bonds and seven I$^-$ ions via Cu-I bonds furnishing the full structure of Au$_{12}$Cu$_{13}$ cluster. In detail, five I anions are coordinated to ten Cu atoms in Cu$_{11}$ unit taking a $\mu_2$-bridging coordination mode (i.e., one I anion per two Cu atoms of Cu$_{11}$ unit with an average Cu-I bond length of ~2.586(2) Å), and the remaining two I anions are bonded to the vertex Cu atoms, Fig. 3c. In another way, the rod-like Au$_{12}$Cu$_{13}$ cluster could be deemed as the fusion of two icosahedron Au$_6$Cu$_7$ units by sharing a Cu vertex like the well-known rod-like M$_{25}$ (M = Au, Ag, Cu) clusters[14,15,32]. Of note, Au$_{12}$Cu$_{13}$ cluster is different from Cu$_x$Au$_{25-x}$(PPh$_3$)$_{10}$(PhC$_2$H$_4$S)$_5$Cl$_2$ with an eclipsed arrangement in structure, where the copper atoms partially occupy the top and M$_{11}$ sites and bonded with chlorine or thiolate ligands[33].

In a typical way, the core structure of Au$_{12}$Cu$_{13}$ and Au$_{25}$ can be regarded as a four-layer cylinder[32], Supplementary Figure 4. It is worth noting that the two neighboring pentagons of layer II and layer III in Au$_{12}$Cu$_{13}$ showed a staggered arrangement and adopt a torsion angle of 12.1° (Fig. 3d). In comparison, no twisting is observed in Au$_{25}$ cluster (Supplementary Figure 4b), indicating that the Cu atoms cause the torsional stress in Au$_{12}$Cu$_{13}$ cluster, which is plausibly attributed to the different atomic radii of Au (~144 pm) and Cu (~128 pm). Both Au$_{12}$Cu$_{13}$ and Au$_{25}$ kernels are protected by triphenylphosphine and halide (I/Br) ligands in similar coordination patterns. The iodide anions and phosphine ligands coordinate selectively to Cu and Au atoms in Au$_{12}$Cu$_{13}$ cluster, respectively, due to the different electro negativities of Cu and Au.

Furthermore, the various M-M, M-P and M-X (M = Au/Cu, and X = I$^-$/Br$^-$) bond distances are summarized in Supplementary Table 2. The Au$_c$-Au$_p$ and Au$_p$-P bond lengths (c and p denote central and peripheral) of Au$_{12}$Cu$_{13}$ and Au$_{25}$ are found comparable. And the average Cu$_c$-Cu$_p$ (2.717(2) Å) and Cu$_p$-Cu$_p$ (2.761(2) Å) bond distances in Cu$_{11}$ unit of Au$_{12}$Cu$_{13}$ were

found to be substantially smaller than the corresponding Au$_c$-Au$_p$ (2.878(2) Å) and Au$_p$-Au$_p$ (2.940(3) Å) in Au$_{25}$ cluster. Interestingly, compared with these structures of Au$_{12}$Cu$_{13}$, Au$_{25}$ and Au$_{13}$Ag$_{12}$ clusters, they all can deemed as two vertex-sharing icosahedrons with a metal center, which further promote us to associate them with the uncomplete-icosahedral Au$_9$ cluster with an Au center, the starting materials as well. In short, the Cu atoms in system not only lead the reconstruction and assemble of Au$_9$ clusters into Au$_{12}$Cu$_{13}$ clusters, but also caused the structural distortion in the Au$_{12}$Cu$_{13}$ clusters.

**Stability of Au$_{12}$Cu$_{13}$ clusters.** The optimum stability of metal nanocluster is essential for their use in various applications, and it is previously reported that the Cu dopants in gold nanoclusters often caused the instability to the alloy clusters[22,34]. Therefore, we examined the stability of Au$_{12}$Cu$_{13}$ *vis-á-vis* that of Au$_{25}$ clusters in solution. As shown in Fig. 4a, Au$_{12}$Cu$_{13}$ in a CH$_2$Cl$_2$ solution give three obvious absorption features at 448, 508 and 655 nm in the range of 400 to 800 nm, which is notably much distinct from those of Au$_{25}$ (419, 470, 526 and ~660 nm), indicating reasonable perturbation of electronic structure upon Cu doping.

The stability of Au$_{12}$Cu$_{13}$ and Au$_{25}$ clusters in CH$_2$Cl$_2$ was evaluated under irradiation of sunlight via time-dependent UV-vis spectroscopy using crystal samples. As depicted in Fig. 4c, UV-vis profile of homo-Au$_{25}$ decreased obviously in 30 min and almost disappeared after 60 min indicating the poor stability of homo-Au$_{25}$ cluster in CH$_2$Cl$_2$. Whereas the profile Fig. 4b assigned to Au$_{12}$Cu$_{13}$ clusters kept intact illustrating Au$_{12}$Cu$_{13}$ cluster is higher stability than corresponding homo-Au$_{25}$ clusters owing to the dopant of Cu atoms. Furthermore, the stability of Cu species in Au$_{12}$Cu$_{13}$ cluster have been improved significantly in solution exposed to air since these UV-Vis profiles in Fig. 4d remained overlapping over 7 hours and no precipitation formed, which is unusual even for Cu$^I$ complexes. Hence the Au$_{12}$Cu$_{13}$ with high stability in solution under light or in air is achieved, which is not observed in other copper doped M$_{25}$ nanoclusters[22]. Although the rod-shaped clusters of Au$_{12}$Cu$_{13}$ and Au$_{25}$ share the similar configuration and a 16-electron system (i.e. the dimer of 8-electron superatom)[35], the composition in core differs suggesting the synergistic effects between Au and Cu may be an important reason for the high stability of Au$_{12}$Cu$_{13}$ cluster. Secondly, the ligated I$^-$ ions with large radius (γ: ~ 2.2 Å) at waist of Au$_{12}$Cu$_{13}$ cluster exhibit higher coordinate capability than Br$^-$ (γ: ~1.96 Å) in Au$_{25}$ cluster.

**Fluorescence property.** We next investigated the fluorescence (FL) property of Au$_{12}$Cu$_{13}$ nanocluster. The excitation curve from 400 nm to 700 nm in Fig. 5 shares the similar profile with UV-vis spectrum of Au$_{12}$Cu$_{13}$ monitored by 774 nm, suggesting the fluorescence of Au$_{12}$Cu$_{13}$ generated from the metal core cluster and different from small metal-organic complexes with similar

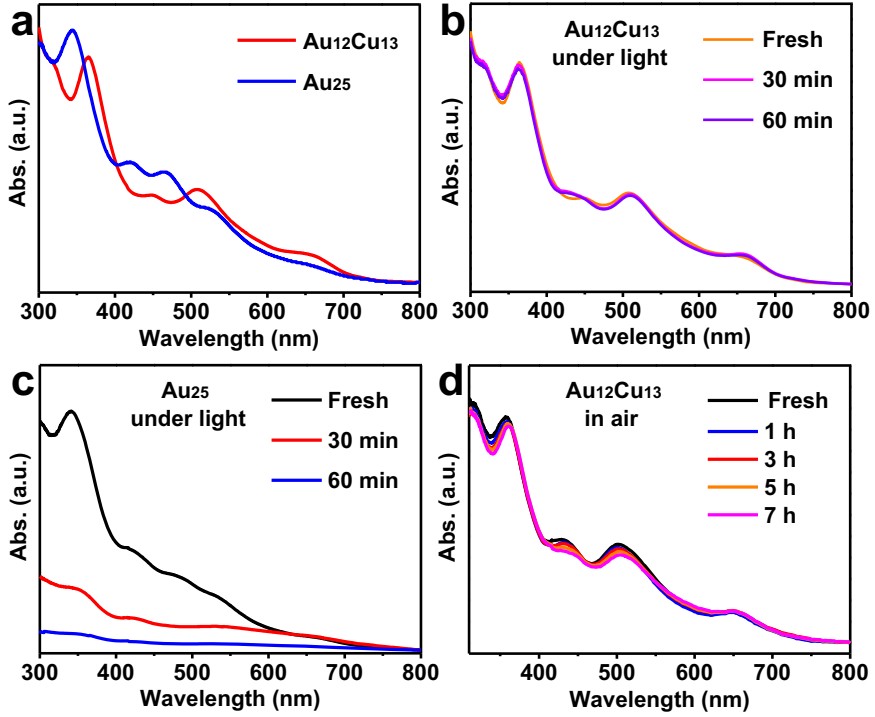

**Fig. 4 Stability of Au₁₂Cu₁₃ clusters. a** UV-vis spectra of Au₁₂Cu₁₃ and Au₂₅ nanoclusters. Stability tests of Au₁₂Cu₁₃ and Au₂₅ clusters: **b**, **c** under sunlight and **d** in air.

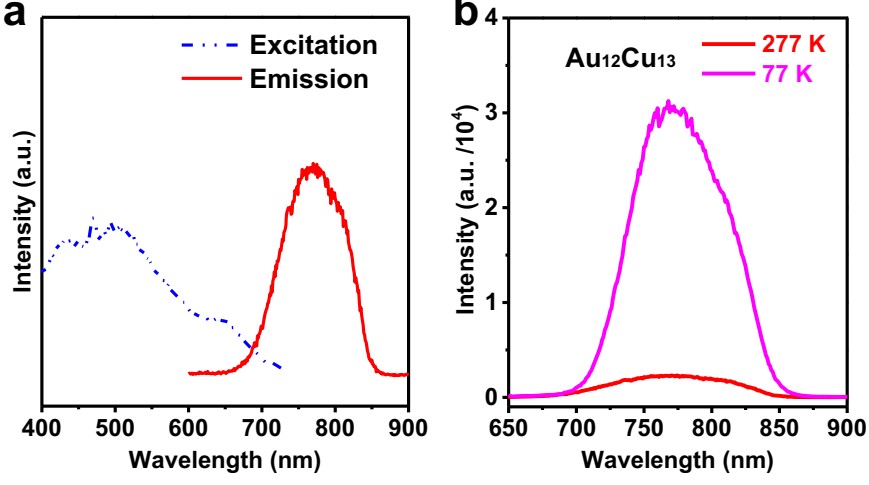

**Fig. 5 Fluorescence property of Au₁₂Cu₁₃ clusters. a** Excitation and emission spectra of Au₁₂Cu₁₃ cluster in solution. **b** Temperature-dependent fluorescence of Au₁₂Cu₁₃ clusters in solution.

ligands, whose fluorescence need to be pumped by UV lower than 400 nm. Interestingly, the intensity of fluorescence has been enhanced significantly once the Cu atoms were doped. An obvious broad fluorescence emission centered at 774 nm was noticed in the wavelength range from 660 to 850 nm, excited at $\lambda_{ex} \sim 470$ nm as depicted in Fig. 5, which is broader than that of Au₂₅ clusters (700–850 nm, Supplementary Figure 5). Further, the maximum emission wavelength of Au₁₂Cu₁₃ at $\lambda_{em} \sim 774$ nm was found red shifted with respect to Au₂₅ clusters ($\lambda_{em} \sim 750$ nm, $\lambda_{ex} \sim 470$ nm) and Au₁₂Ag₁₃ ($\lambda_{em} \sim 739$ nm, $\lambda_{ex} \sim 430$ nm) clusters[14]. Besides, the excitation spectrum monitored at 774 nm emission (Fig. 5a, blue dash line) was found almost identical to the absorption spectrum of Au₁₂Cu₁₃ cluster (Fig. 5a, red line), and Stokes shift of Au₁₂Cu₁₃ cluster was determined as 304 nm close

to that of Ag atoms doped Au₁₂Ag₁₃ cluster (309 nm)[14] and smaller than that of Au₂₅ cluster (330 nm). The quantum yield (QY) of Au₁₂Cu₁₃ was determined to be ~34% by an absolute method, which is significantly higher than Au₁₂Ag₁₃ (QY: ~26%)[14] and Au₂₅ clusters (~1%, by absolute method) and close to the highly fluorescent [Au₁₂Ag₁₃(PPh₃)₁₀(SR)₅Cl₂]²⁺ nanocluster (~40%)[15], where the 13 Ag was doped in the center position similar to our Cu-centered Au₁₂Cu₁₃ cluster.

Furthermore, the FL lifetime of Au₁₂Cu₁₃ is up to 900 ns deduced from Supplementary Figure 6, and an almost linear fading tendency was detected indicating the nature of mono-exponential FL decay dynamics of Au₁₂Cu₁₃ cluster in solution. These results illustrated the synergistic effect of Cu and Au atoms in Au₁₂Cu₁₃ clusters plays a key role in the drastic enhancement

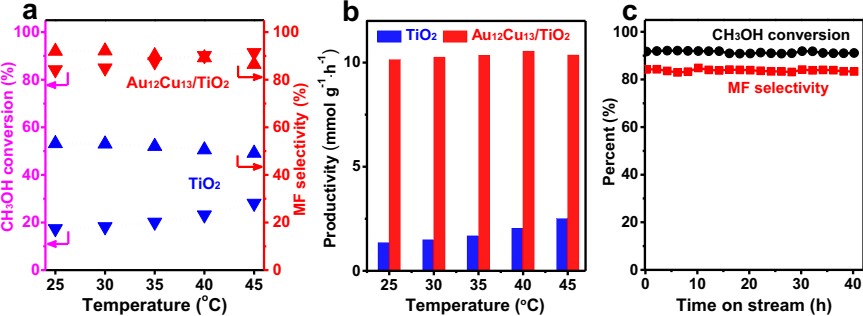

**Fig. 6 Catalytic performance of Au₁₂Cu₁₃/TiO₂ in the photo-oxidation of methanol. a** Catalytic performance as a function of temperature on TiO₂ (P25) and Au₁₂Cu₁₃/TiO₂. **b** Formation rate of methyl formate (MF) over TiO₂ and Au₁₂Cu₁₃/TiO₂ catalysts. **c** Durability test of the Au₁₂Cu₁₃/TiO₂ catalysts at 25 °C. Reaction conditions: ~ 20 mg catalysts, λ = 365 nm, methanol (1.0 v %) and O₂ (0.5 v %) balanced with N₂ at the flow rate of 20 mL min⁻¹.

of fluorescence. We further investigated the temperature-dependent fluoresce (TDFL) of $Au_{12}Cu_{13}$ clusters at 277 and 77 K, Fig. 5b. An obvious enhancement of about 10-fold was achieved when the cluster solution was cooled to 77 K from 277 K, and FL emission kept unchanged, suggesting no-radiative transmittance $Au_{12}Cu_{13}$ at low temperature decreased obviously. TDFL analysis proves that FL of $Au_{12}Cu_{13}$ may be originated from the same excited state and the electron structure of $Au_{12}Cu_{13}$ is not temperature dependent[18].

**Catalytic test in photo-oxidation of methanol**. The structurally well-identified and highly stable $Au_{12}Cu_{13}$ clusters with characteristic absorption features and excellent fluorescence can serve as a promising photocatalyst towards oxidation of methanol to methyl formate as both gold and copper species are considered as excellent candidates for this transformation[14,29,36,37]. Thus, we tested $Au_{12}Cu_{13}$ clusters as catalyst for photo-oxidation of methanol. The catalytic experiment details were given in Experimental section. In brief, ~0.5 wt% of $Au_{12}Cu_{13}$ nanocluster was firstly loaded on surface of $TiO_2$, followed by treatment of $Au_{12}Cu_{13}/TiO_2$ samples with amorphous $Al_2O_3$ using atomic layer deposition (ALD) technique to grow the shell/cage of $Al_2O_3$ (100 ALD cycles) around the $Au_{12}Cu_{13}$ cluster at 150 °C[38], where $Au_{12}Cu_{13}$ clusters keep inert. The photocatalytic results are depicted in Fig. 6 and Supplementary Figure 7. It is worthy to note that no methanol conversion was detected when the light and catalysts were absent, demonstrating that the photo-oxidation occurred over the $Au_{12}Cu_{13}/TiO_2$ catalysts. And it is found that methanol conversion was gradually improved, and methyl formate-selectivity was gradually decreased with the increasing reaction temperatures (from 25 to 45 °C), as shown in Fig. 6a. The highest methyl formate formation rate over $Au_{12}Cu_{13}/TiO_2$ is evaluated to be ~10.6 mmol g⁻¹ h⁻¹ at 40 °C, which is 4-folds of that over $TiO_2$ (Fig. 6b) and substantially higher than previously reported photocatalysts, Supplementary Table 3 [39].

Further, the durability of $Au_{12}Cu_{13}/TiO_2$ catalyst for photooxidation of methanol was tested. Significantly, during the whole process (~40 h), the cluster catalyst gave rise to a constant ~92% methanol conversion and a ~84% methyl formate-selectivity with no appreciable loss of activity, as shown in Fig. 6c. These results strongly indicate the robust nature of the $Au_{12}Cu_{13}$ composite, holding promise of outstanding catalytic activity for the prolonged period of time.

We have developed a strategy to acquire the rod-like $[Au_{12}Cu_{13}(PPh_3)_{10}I_7](SbF_6)_2$ nanocluster, which can also be extended to the preparation of $[Au_{25}(PPh_3)_{10}Br_7](SbF_6)_2$ and $[Au_{25-y}Ag_y(PPh_3)_{10}CI_8](SbF_6)$ nanoclusters. A plausible mechanism for the $Au_{12}Cu_{13}$ formation that $Au_{12}Cu_{13}$ clusters were constructed by the dimerization of metastable intermediates like $[Au_8Cu(Ph_3P)_8]^{2+}$ and $[Au_9Cu(Ph_3P)_8I]^{2+}$ generated from the reaction of $Au_9$ clusters and CuI was presented and demonstrated by UV-vis spectroscopy, ESI-MS and single crystal X-ray diffraction technologies. The obtained $Au_{12}Cu_{13}$ cluster exhibits high stability in solution and outstanding photo-luminescence character with QY ≈ 34%, which laid the foundation for the applications in photoluminescence and photo-catalysis. Furthermore, the $Au_{12}Cu_{13}$ clusters showed good catalytic performance in the photo-oxidation of methanol toward methyl formate and good durability. The present work deepens the understanding of the assembling mechanism of clusters and provide the future guidelines for the highly controllable synthesis of functionalized alloy nanoclusters.

## Methods

**Synthesis of M₂₅ clusters**. Typically, Ph₃PAuCl (25 mg) reacted with AgSbF₆ (17.2 mg) in 4 mL of CH₂CL₂ and methanol (v/v = 1), and the obtained clear solution was reduced by NaBH₄ solution (2 mg dissolved in ice-cold methanol) at 0 °C, giving a dark brown solution. After ~24 h of stirring, CuI (9.5 mg) was added into the mixture and kept stirring at 0 °C. The solution was filtrated when the characteristic peaks at 352, 434, 508 and 655 nm appeared. The filtrate was further evaporated to dryness under vacuum, which was further washed twice with a mixture of hexane and dichloromethane (v/v = 5:1). The pure clusters were finally extracted with a mixed solution of CH₂Cl₂ and CH₃OH (v/v = 1). A black block crystal of Au₁₂Cu₁₃ was obtained via a slow diffusion of diethyl ether into the dichloromethane solution of clusters over two weeks. Yield: 15.25 mg, 50.5% (based on Au). The Au₂₅ (yield: ~58%) and Au₂₅₋ₓAgₓ (yield: ~49%) clusters were obtained through a similar method by using KBr (11 mg, 0.05 mmol) and AgCl (7 mg, 0.05 mmol) to replace CuI, respectively. The detailed characterizations of the clusters are given in in the SI.

**Catalytic performance evaluation**. The Au₁₂Cu₁₃/TiO₂ catalyst was prepared by the method of atomic layer deposition; see details in the SI. The oxidations of methanol to methyl formate were carried out in a home-made continuous-flow aluminum alloy reactor with a rectangle quartz window on the top. A 500 W high-pressure mercury lamp (CEl-LAM 500) with a wavelength of 365 nm was employed as the light source, installed above the quartz window of the reactor with the light intensity of 18.6 mW cm⁻². The reaction temperatures were controlled at 25–40 °C using a cooling water circulation. 20 mg of catalysts were uniformly coated on the glass's surface, which is fully exposed to light irradiation. A gas mixture containing 1.0 v% methanol, 0.5 v% O₂ balanced with N₂ at the flow rate of 20 mL·min⁻¹ was introduced into reactor that is bubbled through a liquid methanol in a flask. The effluent was examined by an on-line Agilent 7820 with a thermal conductivity detector (TCD) and a flame ionization detector (FID).

**Further details**. See Supplementary Methods for details on X-ray crystallographic structural determinations, additional nanocluster characterizations, and preparation of the Au₁₂Cu₁₃/TiO₂ catalyst.

## Data availability

Crystallographic data for [Au₁₂Cu₁₃(Ph₃P)₁₀I₇](SbF₆)₂ and [Au₂₅(Ph₃P)₁₀Br₇](SbF₆)₂ is deposited in The Cambridge Crystallographic Data Centre (CCDC) as CCDC-1965910 and CCDC-1966985 (Supplementary Data 1 and 2). The data supporting the findings of this study are available within this article and its Supplementary Information. Extra data are available from the corresponding author upon reasonable request.

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

## Acknowledgements

We are grateful for the provision of beam time at the BL17B beamline of the National Facility for Protein Science (NFPS), Shanghai Synchrotron Radiation Facility (SSRF) Shanghai, China.

## Author contributions

Z.Q. and G.L. conceived and designed the project. J.Z., Y.Z., Z.Q., and Z.L. carried out the experiments. Z.Q., S.S., and G.L. wrote the manuscript, and all authors analyzed the data and discussed the results.

## Competing interests

The authors declare no competing interests.
