## [Peer Review File · Communications Chemistry]

Reviewers' comments:

Reviewer #1 (Remarks to the Author):

This work demonstrated Cu-doped rod-like alloy nanoclusters with the chemical composition of $[\text{Au}_{12}\text{Cu}_{13}(\text{PPh}_3)_{10}\text{I}7](\text{SbF}_6)_2$, showing high stability and durability. I do not recommend publication in Communications Chemistry due to many mistakes and the lack of experimental results in its current form.

Some issues that need to be addressed:

1) The author mentioned the 8-electron superatom dimer in the abstract but it is not illustrated in the whole manuscript.

2) It should be 933.1 eV rather than 9331 eV in line 130 of page 6.

3) The XPS peak in 932.5 eV is very weak. I do not think it can be simply assigned to be Cu(0). The peak was assigned as Cu(I) in a previous research, Surf. Sci. Spectra 2, 149 (1993). Other characterizations like XANES are recommended adopting as auxiliary evidence. Moreover, hydrides like to exist in the alloy nanocluster in the presence of NaBH_4 .

4) The sentence "the fluorescence of $\text{Au}_{12}\text{Cu}_{13}$ generated from the metal core cluster and different from small metal-organic complexes, whose fluorescence need to be pumped by UV lower than 400 nm" in line 218 of page 9 is not correct. Ligands determine the excitation wavelength of small metal-organic complexes to some degree, and the excitation wavelength of these complexes can be lower than 400 nm as long as the ligand is conjugate enough.

5) It is suggested to provide theoretical calculation results to support the assignment of emission bands.

6) A solid-state emission spectrum should be given.

Reviewer #2 (Remarks to the Author):

In this work, the authors reported the synthesis of rod-like $\text{Au}_{12}\text{Cu}_{13}$ nanoclusters via the transformation of Au_9 Intermediate in the presence of CuI. It was unveiled that Cu-doping in the nanoclusters exert substantial influence on the physicochemical properties of original nanoclusters including photoluminescence and photocatalytic performances. The idea of this work is impressive in terms of material synthesis and catalytic investigation. Moreover, the manuscript is well organized. After carefully reviewing this work, I believe it can be recommended for publication in this journal after minor revision based on the following suggestions.

1. The high-resolution Au 4f spectrum (Figure 2c) should be well deconvoluted to differentiate the Au(I) and Au(0) species.

2. Control experiments without light or catalyst involving the photocatalytic reaction should be performed to evidence the reaction is truly a photocatalytic reaction.

3. Some important works of similar research motif should be considered such as ACS Catal., 2022, 12,

4216-4226. *J. Mater. Chem. A*, 2022, 10, 7006-7012. *J. Mater. Chem. A*, 2022, 10, 4032-4042. *J. Phys. Chem. C*, 2021, 125, 22421-22428.

Reviewer #3 (Remarks to the Author):

In this work, Li, Qin and coworkers reported the synthesis of a rod-like nanocluster with chemical composition of $\text{Au}_{12}\text{Cu}_{13}(\text{PPh}_3)_{10}\text{I}_7$. The Au-Cu alloyed $\text{Au}_{12}\text{Cu}_{13}$ nanocluster was obtained by reacting the $\text{Au}_9(\text{PPh}_3)_8$ nanocluster with CuI, and the transformation was tracked by UV-vis, ESI-MS, and SC-XRD. The Cu-dopants significantly improved the stability and fluorescence of the $\text{Au}_{12}\text{Cu}_{13}$ nanocluster, relative to homogold Au_{25} . The high stability of $\text{Au}_{12}\text{Cu}_{13}$ is attributed to the high binding energy of iodine ligands and the electronic structure as an 8-electron superatom dimer. The catalytic performance of the $\text{Au}_{12}\text{Cu}_{13}$ was also evaluated. The reported preparation, structure determination, and property investigation of the $\text{Au}_{12}\text{Cu}_{13}(\text{PPh}_3)_{10}\text{I}_7$ nanocluster is interesting. I would like to suggest the acceptance of this paper after the authors have addressed the following minor issues.

(A) For Ag-alloyed rod-like $\text{Ag}_x\text{Au}_{25-x}(\text{PPh}_3)_{10}(\text{SR})_5\text{Cl}_2$ nanoclusters, the Ag dopants are always uncertain, such as $\text{Ag}_{13}\text{Au}_{12}$, $\text{Ag}_{12}\text{Au}_{13}$, $\text{Ag}_{11}\text{Au}_{14}$, etc. For the Cu-alloyed $\text{Cu}_x\text{Au}_{25-x}(\text{PPh}_3)_{10}(\text{SR})_5\text{Cl}_2$ nanocluster, the introduced Cu atoms were determined as 13, without any other components. That is interesting. The authors should present some discussion on the difference between the Ag and Cu dopants.

(B) The author demonstrated that "quantum yield: $\sim 34\%$, 34-folds of homo- $\text{Au}_{25}(\text{PPh}_3)_{10}\text{Br}_7$ ". Is the PL QY of $\text{Au}_{25}(\text{PPh}_3)_{10}\text{Br}_7$ also determined by an absolute method?

(C) The current work focuses on the alloying process and the fluorescence of metal nanoclusters. Several related works or reviews should be noted: *Chem. Soc. Rev.*, 2020, 49, 6443; *Nat. Commun.*, 2021, 12, 6186; *PNAS*, 2019, 116, 18834; *J. Am. Chem. Soc.*, 2022, 144, 4845. Besides, in their structural anatomy, the overall structure of $\text{Au}_{12}\text{Cu}_{13}$ was split into $2*\text{Au}_6\text{Cu}_1 + \text{Cu}_{11}$; another structural analytical mode should be noted, $\text{M}_{13} + \text{M}_{13}$ by sharing a vertex Cu atom (see *Coord. Chem. Rev.*, 2019, 394, 1).

(D) Several typos should be noted, e.g., Figure 5a, emission; Table S2, $\text{Au}_{25}(\text{PPh}_3)_{10}\text{Br}_5$.

Responses to the Reviewers and action taken in revision

We appreciate the Reviewers' critical reading of our manuscript and the valuable suggestions. We have revised the manuscript according to their suggestions. The Reviewers' comments are presented in black and our responses and the actions taken in the revision are in blue typeface.

Response to Reviewers' comments

Reviewer #1

This work demonstrated Cu-doped rod-like alloy nanoclusters with the chemical composition of $[\text{Au}_{12}\text{Cu}_{13}(\text{PPh}_3)_{10}\text{I}_7](\text{SbF}_6)_2$, showing high stability and durability. I do not recommend publication in Communications Chemistry due to many mistakes and the lack of experimental results in its current form.

1) The author mentioned the 8-electron superatom dimer in the abstract but it is not illustrated in the whole manuscript.

Response: Thanks for Reviewer' suggestion. We have illustrated the 8-electron superatom dimer in the revised manuscript.

2) It should be 933.1 eV rather than 9331 eV in line 130 of page 6.

Response: It is our mistake. It has been changed.

3) The XPS peak in 932.5 eV is very weak. I do not think it can be simply assigned to be Cu(0). The peak was assigned as Cu(I) in a previous research, Surf. Sci. Spectra 2, 149 (1993). Other characterizations like XANES are recommended adopting as auxiliary evidence. Moreover, hydrides like to exist in the alloy nanocluster in the presence of NaBH_4 .

Response: Thanks for Reviewer' suggestion. The split of Cu 2p XPS peak was accorded to the previous research, *J. Phys. Chem. C* **112**, 1101-1108 (2008), and the crystal structure of $\text{Au}_{12}\text{Cu}_{13}$ cluster. Further, the auger electron spectroscopy in Figure S2 was also provided in SI to support our judgment.

Besides, XANES has been applied in our system, but the signal is too weak to distinguish.

“And the Cu^0 species, generally, should be attributed to the shared vertex Cu atom, while

the Cu^I one should belong to the peripheral and end vertex Cu atoms bonded directly with I atoms according to SCXRD analysis (*vide infra*)”

4) The sentence “the fluorescence of Au₁₂Cu₁₃ generated from the metal core cluster and different from small metal-organic complexes, whose fluorescence need to be pumped by UV lower than 400 nm” in line 218 of page 9 is not correct. Ligands determine the excitation wavelength of small metal-organic complexes to some degree, and the excitation wavelength of these complexes can be lower than 400 nm as long as the ligand is conjugate enough.

Response: Thanks for Reviewer’ suggestion. We have changed this sentence as following: “The excitation curve from 400 nm to 700 nm in Fig. 6 shares the similar profile with UV-vis spectrum of Au₁₂Cu₁₃ monitored by 774 nm, suggesting the fluorescence of Au₁₂Cu₁₃ generated from the metal core cluster and different from small metal-organic complexes with similar ligands, whose fluorescence need to be pumped by UV lower than 400 nm.”

5) It is suggested to provide theoretical calculation results to support the assignment of emission bands.

Response: Thanks for Reviewer’ suggestion. We fully agree with Reviewer’ point that the theoretical calculation will be helpful to the assignment of emission bands. And we have tried to carry out the DFT, but unfortunately, no useful results were obtained for ~ two months.

6) A solid-state emission spectrum should be given.

Response: Thanks for Reviewer’s suggestion. Our crystal sample is not enough for the measurement of solid-state emission spectrum.

Reviewer #2

In this work, the authors reported the synthesis of rod-like Au₁₂Cu₁₃ nanoclusters via the transformation of Au₉ Intermediate in the presence of CuI. It was unveiled that Cu-doping

in the nanoclusters exert substantial influence on the physicochemical properties of original nanoclusters including photoluminescence and photocatalytic performances. The idea of this work is impressive in terms of material synthesis and catalytic investigation. Moreover, the manuscript is well organized. After carefully reviewing this work, I believe it can be recommended for publication in this journal after minor revision based on the following suggestions.

1. The high-resolution Au 4f spectrum (Figure 2c) should be well deconvoluted to differentiate the Au(I) and Au(0) species.

Response: Thanks for Reviewer' kind suggestion. It is done and added as new Fig. 2c.

2. Control experiments without light or catalyst involving the photocatalytic reaction should be performed to evidence the reaction is truly a photocatalytic reaction.

Response: Thanks for Reviewer' kind suggestion. No methanol conversion was detected when the light and catalysts were absent. And the discussion was added:

“It is worthy to note that no methanol conversion was detected when the light and catalysts were absent, demonstrating that the photo-oxidation occurred over the Au₁₂Cu₁₃/TiO₂ catalysts.”

3. Some important works of similar research motif should be considered such as ACS Catal., 2022, 12, 4216-4226. J. Mater. Chem. A, 2022, 10, 7006-7012. J. Mater. Chem. A, 2022, 10, 4032-4042. J. Phys. Chem. C, 2021, 125, 22421–22428.

Response: The related literatures have been cited.

Reviewer #3

In this work, Li, Qin and coworkers reported the synthesis of a rod-like nanocluster with chemical composition of Au₁₂Cu₁₃(PPh₃)₁₀I₇. The Au-Cu alloyed Au₁₂Cu₁₃ nanocluster was obtained by reacting the Au₉(PPh₃)₈ nanocluster with CuI, and the transformation was tracked by UV-vis, ESI-MS, and SC-XRD. The Cu-dopants significantly improved the stability and fluorescence of the Au₁₂Cu₁₃ nanocluster, relative to homo-gold Au₂₅. The high

stability of $\text{Au}_{12}\text{Cu}_{13}$ is attributed to the high binding energy of iodine ligands and the electronic structure as an 8-electron superatom dimer. The catalytic performance of the $\text{Au}_{12}\text{Cu}_{13}$ was also evaluated. The reported preparation, structure determination, and property investigation of the $\text{Au}_{12}\text{Cu}_{13}(\text{PPh}_3)_{10}\text{I}_7$ nanocluster is interesting. I would like to suggest the acceptance of this paper after the authors have addressed the following minor issues.

(A) For Ag-alloyed rod-like $\text{Ag}_x\text{Au}_{25-x}(\text{PPh}_3)_{10}(\text{SR})_5\text{Cl}_2$ nanoclusters, the Ag dopants are always uncertain, such as $\text{Ag}_{13}\text{Au}_{12}$, $\text{Ag}_{12}\text{Au}_{13}$, $\text{Ag}_{11}\text{Au}_{14}$, etc. For the Cu-alloyed $\text{Cu}_x\text{Au}_{25-x}(\text{PPh}_3)_{10}(\text{SR})_5\text{Cl}_2$ nanocluster, the introduced Cu atoms were determined as 13, without any other components. That is interesting. The authors should present some discussion on the difference between the Ag and Cu dopants.

Response: Thanks for Reviewer' suggestion. We have added the discussion on the difference between the Ag and Cu dopants.

(B) The author demonstrated that “quantum yield: $\sim 34\%$, 34-folds of homo- $\text{Au}_{25}(\text{PPh}_3)_{10}\text{Br}_7$ ”. Is the PL QY of $\text{Au}_{25}(\text{PPh}_3)_{10}\text{Br}_7$ also determined by an absolute method?

Response: Thanks for Reviewer' suggestion. The QY of $\text{Au}_{25}(\text{PPh}_3)_{10}\text{Br}_7$ was determined by an absolute method.

(C) The current work focuses on the alloying process and the fluorescence of metal nanoclusters. Several related works or reviews should be noted: Chem. Soc. Rev., 2020, 49, 6443; Nat. Commun., 2021, 12, 6186; PNAS, 2019, 116, 18834; J. Am. Chem. Soc., 2022, 144, 4845. Besides, in their structural anatomy, the overall structure of $\text{Au}_{12}\text{Cu}_{13}$ was split into $2*\text{Au}_6\text{Cu}_1 + \text{Cu}_{11}$; another structural analytical mode should be noted, $\text{M}_{13} + \text{M}_{13}$ by sharing a vertex Cu atom (see Coord. Chem. Rev., 2019, 394, 1).

Response: Response: The related literatures have been cited.

(D) Several typos should be noted, e.g., Figure 5a, emission; Table S2, $\text{Au}_{25}(\text{PPh}_3)_{10}\text{Br}_5$.

Response: Thanks for Reviewer's kind suggestion. These errors have been corrected in the revised version.

REVIEWERS' COMMENTS:

Reviewer #1 (Remarks to the Author):

I am fine with the revision. The manuscript is acceptable in its current form.

Reviewer #2 (Remarks to the Author):

The quality of this work has been improved and now it can be published as it is.

Reviewer #3 (Remarks to the Author):

The points raised in the previous round of review have been satisfactorily addressed. This work could be accepted in its current form.